# Comparison of IMU-Based Knee Kinematics with and without Harness Fixation against an Optical Marker-Based System

**DOI:** 10.3390/bioengineering11100976

**Published:** 2024-09-28

**Authors:** Jana G. Weber, Ariana Ortigas-Vásquez, Adrian Sauer, Ingrid Dupraz, Michael Utz, Allan Maas, Thomas M. Grupp

**Affiliations:** 1Research and Development, Aesculap AG, 78532 Tuttlingen, Germanythomas.grupp@aesculap.de (T.M.G.); 2Department of Orthopaedic and Trauma Surgery, Musculoskeletal University Center Munich (MUM), Campus Grosshadern, Ludwig Maximilians University Munich, 81377 Munich, Germany

**Keywords:** IMU, gait analysis, knee kinematics, REFRAME, motion capture, movement biomechanics, wearables

## Abstract

The use of inertial measurement units (IMUs) as an alternative to optical marker-based systems has the potential to make gait analysis part of the clinical standard of care. Previously, an IMU-based system leveraging Rauch–Tung–Striebel smoothing to estimate knee angles was assessed using a six-degrees-of-freedom joint simulator. In a clinical setting, however, accurately measuring abduction/adduction and external/internal rotation of the knee joint is particularly challenging, especially in the presence of soft tissue artefacts. In this study, the in vivo IMU-based joint angles of 40 asymptomatic knees were assessed during level walking, under two distinct sensor placement configurations: (1) IMUs fixed to a rigid harness, and (2) IMUs mounted on the skin using elastic hook-and-loop bands (from here on referred to as “skin-mounted IMUs”). Estimates were compared against values obtained from a harness-mounted optical marker-based system. The comparison of these three sets of kinematic signals (IMUs on harness, IMUs on skin, and optical markers on harness) was performed before and after implementation of a REference FRame Alignment MEthod (REFRAME) to account for the effects of differences in coordinate system orientations. Prior to the implementation of REFRAME, in comparison to optical estimates, skin-mounted IMU-based angles displayed mean root-mean-square errors (RMSEs) up to 6.5°, while mean RMSEs for angles based on harness-mounted IMUs peaked at 5.1°. After REFRAME implementation, peak mean RMSEs were reduced to 4.1°, and 1.5°, respectively. The negligible differences between harness-mounted IMUs and the optical system after REFRAME revealed that the IMU-based system was capable of capturing the same underlying motion pattern as the optical reference. In contrast, obvious differences between the skin-mounted IMUs and the optical reference indicated that the use of a harness led to fundamentally different joint motion being measured, even after accounting for reference frame misalignments. Fluctuations in the kinematic signals associated with harness use suggested the rigid device oscillated upon heel strike, likely due to inertial effects from its additional mass. Our study proposes that optical systems can be successfully replaced by more cost-effective IMUs with similar accuracy, but further investigation (especially in vivo and upon heel strike) against moving videofluoroscopy is recommended.

## 1. Introduction

Gait analysis is the systematic study of locomotion of human legs during gait and can be used in both clinical and research settings, such as patient diagnostics and biomechanical studies. One of the dominant state-of-the-art technologies is optical marker-based motion capture (OMC) systems. Kinematic signals are determined computationally based on the tracked positions of reflective markers in three-dimensional space using multiple infrared cameras [1]. Maintaining consistent environmental conditions is important, as factors such as lighting and reflective surfaces can affect the accuracy of dynamic measurements, so setup typically requires an expert and is time-consuming. These characteristics tend to tie OMC systems to laboratory conditions and associate them with high costs [2].

In recent years, the field of gait analysis has witnessed a significant shift towards the development of more mobile solutions with the emergence of wearable technologies such as inertial measurement units (IMUs). These systems offer a promising alternative to traditional optical marker-based approaches for assessing human movement patterns, particularly in the context of measuring rotational knee kinematics [3,4,5,6]. IMU-based systems are associated with more affordable prices, are more user-friendly, have smaller dimensions, and are therefore particularly well-suited for capturing motion inside and outside of laboratory settings [6,7]. Their application usually relies on processing linear acceleration data collected using an accelerometer, and angular velocity data collected using a gyroscope, to then estimate joint kinematics using sensor fusion algorithms. One of the key benefits of IMU-based systems is thus their potential to make gait analysis more accessible and cost-effective, enabling their use outside of laboratory settings, potentially even as part of routine clinical patient pathways.

Prior to clinical application, any gait analysis system should be thoroughly validated [8]. To that end, the accuracy of an IMU-based system to estimate knee joint angles was previously assessed using a six-degrees-of-freedom joint simulator [9]. Guided by fluoroscopy-based signals originally captured in vivo, the simulator replicated the tibiofemoral motion of six total knee arthroplasty patients performing daily activities [10], while excluding the possible influence of soft tissue artefact. Raw inertial data collected using a pair of IMUs attached to the simulator were then processed using a Rauch–Tung–Striebel smoother to derive flexion/extension, abduction/adduction, and internal/external tibial rotation based on IMU measurements. To evaluate differences between the simulator and IMU-based signals, the root-mean-square errors (RMSEs) between them were calculated. For level walking, results showed mean RMSEs of 0.7° ± 0.1° for flexion/extension, 0.6° ± 0.3° for abduction/adduction, and 0.9° ± 0.2° for external/internal rotation, indicating promising accuracy. Another study tested the algorithm against an optical marker-based system with single iterations of a loaded squat cycle simulated on seven cadaveric specimens on a force-controlled knee rig and achieved mean RMSE values after aligning the underlying reference frames of 4.2° ± 3.6°, 0.9° ± 0.4°, and 1.5° ± 0.7° for flexion/extension, abduction/adduction, and external/internal rotation, respectively [11]. In a clinical setting, however, motion capture systems consisting of skin-mounted sensors (whether optical or inertial) are usually subject to errors caused by non-rigid movements of the skin (and other soft tissue) relative to the underlying bone. This phenomenon, known as soft tissue artefact (STA), can involve either (a) the collective displacement of a group of markers, or (b) the variation in individual inter-marker distances due to skin elasticity [12,13].

Consequently, a next step in the analytical validation of the described IMU-based knee kinematics analysis system is therefore to utilize the system in vivo, where results may be affected by errors due to STA. In the following study, we present an in vivo examination of IMU-based tibiofemoral kinematic estimates, considering two distinct configurations of IMU placement: (1) IMUs attached to a rigid harness (referred to as “IMUs on harness”, i.e., with harness fixation), and (2) IMUs mounted “on the skin” using elastic hook-and-loop bands (referred to as “IMUs on skin”, i.e., without harness fixation). (Note that “IMUs on skin” were technically not directly adhered onto the skin). Both datasets were then compared against a reference signal calculated from optical markers attached to the rigid harness (“OMC on harness”, i.e., the KneeKG system) [14,15,16]. The underlying sensor fusion algorithm used to calculate the IMU-based knee joint angle estimates presented here was based on Ortigas-Vásquez et al.’s adaptation [9] of the approaches previously developed by Seel et al. [3] and Versteyhe et al. [17]. IMU-based knee joint angles were estimated using the previously tested adaptation of Rauch–Tung–Striebel smoothing, after which the REference FRame Alignment MEthod (REFRAME) [18,19] was implemented to account for differences in coordinate system orientations, thus enabling a more rigorous comparison between the three sets of kinematic data (IMUs on harness, IMUs on skin, OMC on harness). Previous studies implementing the REFRAME approach on IMU data tested either on a robotic joint simulator [9] or on cadaveric specimens [11]. This is thus the first study to account for potential differences in local segment reference frame orientations using an optimization-based approach such as REFRAME within an in vivo evaluation of IMU-based tibiofemoral kinematics against optical motion capture.

## 2. Materials and Methods

### 2.1. Participants

The study was conducted according to the guidelines of the Declaration of Helsinki, and approved by the Institutional Review Board (or Ethics Committee) of the medical faculty at Ludwig Maximilian University Munich. Thirty volunteers participated in the study after screening for exclusion criteria (Table 1). After exclusion of all incomplete trials due to (a) equipment failure or (b) missing or corrupted data files, a total of 40 individual knees were considered in the final analysis (sex: 20 female, 20 male; side: 20 right, 20 left; mean age: 31.0 ± 9.6 years; mean height: 1.74 ± 0.07 m; mean body mass: 71.9 ± 13.1 kg; mean body mass index (BMI): 23.7 ± 3.4 $\frac{\mathrm{kg}}{m^{2}}$; mean selected walking speed: 3.9 ± 0.7 $\frac{\mathrm{km}}{h}$).

### 2.2. Reference Motion Capture System

As a reference system, a non-invasive and marker-based OMC system was used (KneeKG, Emovi Inc., Laval, QC, Canada). The system consisted of an infrared camera (Polaris Spectra camera, Northern Digital Inc., Waterloo, ON, Canada), a personal computer equipped with dedicated software (Knee3D, v5.20.7 and v5.20.8, Emovi Inc., Laval, QC, Canada), and passive markers attached to a rigid harness [20]. The computational method employed to define bone-embedded anatomical frames from marker positions and thus estimate knee joint angles has been previously described in detail by Hagemeister et al. [15]. Briefly, the calibration method used by the optical reference system combined the identification of anatomical landmarks and functional calibration tasks. Specifically, during a static trial, the medial and lateral malleoli, and the medial and lateral epicondyles were identified using a pointer (which was itself equipped with optical markers that could be tracked by the cameras). Additionally, participants were asked to perform functional calibration trials to determine the hip and knee joint centers, as well as their postural alignment during neutral standing. In an attempt to minimize STA, the system’s manufacturer trains users to strategically position the provided rigid harness on selected anatomical landmarks [14,20]. Flexion/extension, abduction/adduction, and external/internal rotation values measured by the system have been previously validated at discrete intervals of knee flexion during a quasi-static weight-bearing squatting activity against radiographic images [16]. To the authors’ knowledge, a comprehensive validation of the system during treadmill walking (the system’s intended use) against fluoroscopic imaging is not yet available in the literature. Although the extent to which the device minimizes STA during level walking has therefore yet to be directly assessed, multiple studies that utilize the optical reference system as a validated clinical gait analysis system or “silver standard” have been previously published [21,22,23]. Further studies by, e.g., Lustig et al. [20], Clement et al. [16], and Northon et al. [24], commenting on the validity of the KneeKG system are also available for review. Mean repeatability values have similarly been reported by Hagemeister et al. [15] as ranging between 0.4° and 0.8° for joint rotations, although these values are expected to be highly optimistic and representative of a best-case scenario.

### 2.3. Study Protocol

In addition to total body height and mass, the individual circumferences of each participant’s hip, waist, neck, thighs, and shanks were in turn measured and recorded. Prior to the acquisition of kinematic gait data, each participant underwent a familiarization trial at a self-selected walking speed (0.5 to 6 $\frac{\mathrm{km}}{h}$) in light-colored socks (as indicated by the optical system’s manufacturer). The trial order (right knee assessed first vs. left knee assessed first) was randomized for each participant. The KneeKG marker system (“OMC on harness”) was carefully positioned on each participant by a certified technician (as instructed during official training by the manufacturer of the KneeKG). Additionally, two IMU pairs (Xsens DOT, Movella, Enschede, Netherlands) were attached to the same leg, with one pair of sensors (“IMUs on harness“) adhered to the optical reference’s rigid thigh and shank harness components using double-sided mounting tape, and the other pair (“IMUs on skin“) placed facing (approximately) anterior using elastic hook-and-loop straps secured around the thigh and shank circumferences (Figure 1).

All four IMUs were time-synchronized immediately prior to data collection and set to a sampling rate of 60 Hz. Participants were instructed to begin each trial by spending a minimum of three seconds in a static neutral standing pose, with both knees in full extension and feet facing a direction parallel to the treadmill belt. Participants were instructed to then start the treadmill and set it to increase to the self-selected speed they identified during the preliminary familiarization trial. A few seconds after the chosen walking speed had been reached, data were collected with the OMC system for 45 s. A second set of OMC data was then collected shortly after the first, for another 45 s. Participants then turned off the treadmill, progressively slowing their pace to reach a full stop, and finally adopted the initial neutral reference pose once more, for a minimum of three seconds. At this point, data collection with the IMUs stopped and the entire procedure was repeated for the contralateral knee.

### 2.4. Data Analysis

For each knee, three initial sets of “raw” (non-optimized) kinematic signals were first considered: two sets stemming from the two IMU configurations (IMUs on harness, IMUs on skin), and a third set from the reference system (OMC on harness). The underlying sensor fusion algorithm used for analysis of the IMU data was based on Ortigas-Vásquez et al.’s adaptation [9] of the approaches previously developed by Seel et al. [3] and Versteyhe et al. [17], which leverage Rauch–Tung–Striebel smoothing [25]. This iterative method relied first on a hinge joint model to find an optimal axis of rotation, followed by a ball-and-socket joint model to determine an optimal joint center. The periods of static neutral standing at the beginning and end of each trial were used during post-processing as a reference pose to calibrate any possible offsets in the kinematic signals. For further details, the reader is referred to [3,9,17]. This allowed calculation of knee joint angles from the raw data sampled by each pair of IMUs, using custom Matlab scripts (vR2021b; The Mathworks Inc., Natick, MA, USA). In contrast, the OMC system used built-in software to automatically output joint angle estimates following an approach developed by the manufacturer [15]. Due to the proprietary nature of the software, details regarding, e.g., how the raw marker data were filtered for processing, were not readily available, although previous studies implementing the same harness system have described the use of a zero-lag 2nd-order Butterworth filter with automatically calculated cut-off frequencies [26,27,28]. Each of the resulting sets of kinematic signals consisted of the three rotational kinematic values of the tibiofemoral joint throughout the gait cycle, i.e., flexion/extension, abduction/adduction, and external/internal tibial rotation. Values corresponded to the intrinsic XYZ Cardan angle sequence that transformed the femoral frame into the tibial frame at each time point, where local axis directions were pointed laterally for a right knee for X, anteriorly for Y, and proximally for Z (the resulting joint angles are comparable to those described by Grood and Suntay [29,30]).

IMU- and OMC-based signals were time-synchronized relative to each other by estimating the delay between the OMC-based flexion/extension signal with respect to the IMUs on harness flexion/extension signal (using cross-correlation). The estimated delay was then consistently corrected on all three OMC-based joint angles to collectively align them in time with the IMU-based estimates. (The IMUs on skin and IMUs on harness had already been time-synchronized prior to data collection using dedicated software, as described in Section 2.3). Prior to the implementation of a gait detection algorithm for the identification of individual gait cycles, raw angular velocity values in the sagittal plane of the shank IMU were filtered using a fourth-order Butterworth filter with cut-off frequency at 7 Hz. The negative peaks of the filtered signal were then used for heel strike detection and thus identified the start and end of each stride [31]. The period of every individual cycle was calculated, and those with a duration below the 5th or above the 95th percentile were excluded from further analyses, leaving at least ten valid gait cycles per knee. Each of these 400 gait cycles (40 knees * 10 cycles) were then time-normalized from 0% to 100%.

In order to ensure consistent orientations of local segment coordinate systems across the three sets of kinematic signals, the REference FRame Alignment MEthod (REFRAME) [18] was applied to the normalized cycles. Two different implementations of REFRAME were explored. The first, REFRAME_*IMU*→*OMC*_, minimized the root-mean-square error (RMSE) between each set of IMU-based kinematic signals (IMUs on harness, IMUs on skin) and the OMC system’s raw signals (OMC on harness) as reference. A second implementation, REFRAME_*RMS*_, minimized the root-mean-square (RMS) of abduction/adduction and external/internal rotation of each of the three signal sets (IMUs on harness, IMUs on skin, and OMC on harness) independently. Note that since frame transformations applied by REFRAME were constant across the entire activity cycle, the relative motion between limb segments actually remained the same (it was just illustrated differently). This self-contained implementation additionally minimized the absolute value of flexion/extension at the first timepoint (i.e., 0%) of the cycle. This combination of objective criteria would minimise cross-talk effects, while still allowing for signal convergence with an independent methodology. For both REFRAME_*IMU*→*OMC*_ and REFRAME_*RMS*_, optimizations were formulated to hinder femoral frame transformations consisting of rotations around the femoral X-axis, thus preventing unrealistic changes in the pitch of the femoral and tibial segment frames that could impair the clinical interpretability of the optimized signals. Finally, the RMSEs for each individual gait cycle of the pairwise comparisons between the IMUs (IMUs on harness or IMUs on skin) and the OMC on harness were calculated for each gait cycle before vs. after each REFRAME implementation. Furthermore, the mean RMSEs ± standard deviation across all knees and cycles, and the mean RMSEs ± standard deviation across all cycles of each individual knee were calculated as well.

### 2.5. Statistical Analysis

To evaluate the statistical significance of the mean RMSE differences for each knee before (raw) vs. after REFRAME (REFRAME_*IMU*→*OMC*_ or REFRAME_*RMS*_), a two-tailed paired *t*-test with a significance level of α = 0.05 was performed after each of the two REFRAME implementations independently. Given the considered sample size of 40 knees, testing for normality was not necessary (normality assumption can be violated for *n* > 30 [32]). To account for the possible effects of multiple comparisons, the significance threshold was adapted from 0.05 to 0.004 (2 sensor configurations * 2 REFRAME implementations * 3 planes; therefore, n = 12) using a Bonferroni correction [33].

## 3. Results

For the mean across all knees and cycles, minor differences between the raw kinematic signals (Figure 2, left column) from the IMUs on harness (purple) were observed versus the raw signals from OMC on harness (blue), in the frontal and transverse planes especially. Mean RMSEs between the two systems were 3.8° ± 2.6°, 3.0° ± 2.1°, and 5.1° ± 2.7°, for flexion/extension, abduction/adduction and external/internal rotation, respectively (Table 2). Visibly larger differences could be observed between the IMUs on skin (green) and OMC on harness (blue) (also particularly for abduction/adduction and external/internal rotation) than between the IMUs on harness (purple) and OMC on harness (blue). Mean RMSEs between raw OMC on harness signals and raw IMUs on skin signals were 4.8° ± 2.8°, 3.9° ± 2.1°, and 6.5° ± 2.5°, for flexion/extension, abduction/adduction, and external/internal rotation, respectively (Table 2). Analogous differences were more pronounced in participant-specific results (Figure 3, left column), where mean values were affected by much smaller standard deviations (Table 3). Furthermore, OMC on harness kinematic signals showed clear fluctuations upon heel strike for several knees (e.g., Figure 3; for more examples, see Appendix A). These fluctuations appeared highly repeatable, as demonstrated by the relatively small standard deviation across the trials (e.g., Table 3; for more examples see Appendix A).

The application of REFRAME_*IMU*→*OMC*_ (Figure 2, middle column) resulted in a decrease in mean RMSEs between the OMC on harness (blue) and the IMUs on harness (purple) from 3.8° to 1.1° for flexion/extension, from 3.0° to 0.6° for abduction/adduction, and from 5.1° to 0.9° for external/internal rotation (Table 2). However, mean RMSEs between OMC on harness (blue) and IMUs on skin (green) after the application of REFRAME_*IMU*→*OMC*_ resulted in a comparably smaller decrease from 4.8° to 1.9° for flexion/extension, from 3.9° to 1.5° for abduction/adduction, and from 6.5° to 4.1° for external/internal rotation (Figure 4). On average, the frame transformations executed under REFRAME_*IMU*→*OMC*_ consisted of rotating the local segment frames of the IMUs on harness by no more than 3° around any of the three axes (Table 4). In contrast, the average frame transformations resulting from REFRAME_*IMU*→*OMC*_ for the local segment frames of the IMUs on skin were comparably larger, in some cases exceeding 5°. Notably, the large magnitude of standard deviations affecting these average transformations (Table 5) was not present for participant-specific averages, where mean transformations were affected by much smaller standard deviations (Appendix A, e.g., Appendix A).

Analogously, the implementation of REFRAME_*RMS*_ (Figure 2, right column) resulted in a decrease in mean RMSEs between OMC on harness (blue) and the IMUs on harness (purple) from 3.8° to 1.5° for flexion/extension, from 3.0° to 0.7° for abduction/adduction, and from 5.1° to 0.9° for external/internal rotation (Table 2). Application of REFRAME_*RMS*_ also led to a relatively smaller decrease in mean RMSEs between OMC on harness (blue) and the IMUs on skin (green), from 4.8° to 2.7° for flexion/extension, from 3.9° to 1.7° for abduction/adduction, and from 6.5° to 4.0° for external/internal rotation. Notably, all changes in mean RMSEs were found to be statistically significant even after Bonferroni correction (Figure 4; Appendix A). Similar results were once again more evident for knee-specific averages (Figure 3, right column; Table 3). The transformations executed under REFRAME_*RMS*_ were lowest for OMC on harness and IMUs on harness, compared to those applied to IMUs on skin (Table 6). Once again, participant-specific transformations demonstrated much smaller standard deviations (e.g., Table 7, Appendix A) than inter-participant averages.

## 4. Discussion

Despite showing promising results [3,4,5,31,34], further validation of IMU-based motion capture systems is needed before they are widely used in clinics. Previously [9], a sensor fusion algorithm based on Rauch–Tung–Striebel smoothing [17] was assessed under ideal conditions (i.e., in the absence of STA) using a robotic simulator. In this study, we tested the same IMU-based system in vivo, under the presence of STA, against an optical harness-based reference system.

Both IMU- and optical marker-based motion capture systems aim to characterize the relative motion of the underlying bony segments. To achieve this non-invasively, the systems track (optical or inertial) sensors attached (non-rigidly) to the relevant segments. Any displacement of the sensors relative to the underlying bones therefore results in motion artefacts, which in this context referred not only to STA caused by the movement of skin, muscle, etc., but also displacements of the sensor fixation device (e.g., harness). Here, the IMU system was tested in two distinct configurations (attached to a rigid harness or, alternatively, to the skin using elastic hook-and-loop straps; Figure 1), and differences between the resulting knee kinematic signals were evaluated against a harness-based optical reference system.

A preliminary assessment of the raw tibiofemoral kinematic signals obtained using each IMU configuration revealed visible differences against the optical reference. These differences could stem from a number of sources, especially STA and/or cross-talk effects. In the previous study [9], the IMU-based system characterized rotational knee kinematics during simulated level walking with less than 1° of error in the absence of STA. Given our cohort’s normal average BMI of 23.7 $\frac{\mathrm{kg}}{m^{2}}$, STA magnitudes could reasonably be expected to reach 5° [35,36]. Although average RMSE values between IMUs on skin and the optical reference were roughly within that range, peak RMSEs clearly exceeded these values (>10°, Appendix A), indicating that error sources beyond STA could be present. For IMUs on harness vs. OMC on harness, motion artefacts between systems could even be excluded, as both inertial sensors and optical markers were fixed to the same rigid brace, so peak RMSEs over 10° suggested that cross-talk effects were likely also present.

As described by Hagemeister et al., the optical reference system leveraged joint axes that involved some level of functional calibration, but were still largely dependent on the manual identification of the 3D coordinates of key anatomical landmarks in a laboratory reference frame [15]. On the other hand, the joint axes implemented by the IMU-based system were based only on the available gyroscope and accelerometer data (as the position/orientation of bony points relative to the sensor positions/orientations were not directly measured) and were thus entirely functional [3,9,17]. The joint axes estimated by the different systems were therefore likely similar, but not perfectly coincident [37,38]. Previous work has demonstrated that even very minor differences in joint axis definition leads to considerable artefacts on kinematic signals [18,37,38,39]. We therefore re-assessed signal differences after two distinct implementations of a REference FRame Alignment MEthod (REFRAME) [18] to address such discrepancies in joint axis definition.

In order to address differences in joint axis orientations, REFRAME_*IMU*→*OMC*_ was applied to the set of kinematic signals stemming from each of the two IMU configurations (on harness and on skin) in turn. The underlying goal of this first REFRAME implementation was to re-align the IMU-based local segment reference frames to minimize the RMSE between IMU-based joint angles against the optical reference. The required transformations ranged from as little as 0.0° to as much as 31.0° (Appendix A). In addition to a visible improvement in signal convergence, average RMSEs decreased to well below 2° for the IMUs on harness (Table 2). The level of agreement observed after optimization suggested the underlying motion captured by both systems was highly comparable. Both the optical markers and the IMU sensors on the harness were effectively rigid relative to each other, so after frame alignment, the resulting signals were highly similar. However, average RMSEs between OMC on harness and IMUs on skin after REFRAME_*IMU*→*OMC*_ were higher (up to 5°; Table 2), and clear disagreement remained between signals from the harness-mounted configurations and the skin-mounted IMU set up (Figure 2, middle column; Figure 3, middle column). The remaining differences could not plausibly stem from differences in frame alignment, so alternative sources of error, such as noise and/or measurement error, but especially differences in STA behavior, were suspected. The differences in these kinematic signals therefore indicated that tibiofemoral motion, as characterized by a harness-based system, was inherently different to that captured by a system using skin-mounted sensors. In contrast to the IMUs on harness, the IMUs on skin quantified the motion of the elastic hook-and-loop bands wrapped tightly around the limb segments (rather than the motion of the harness). Since the harness likely moved relative to the elastic hook-and-loop bands, even after reference frame alignment, the IMUs on skin and IMUs on harness did not quantify the same underlying movement pattern, and so those signals remained visibly different.

Even though the implementation of REFRAME_*IMU*→*OMC*_ revealed valuable new insights by characterizing the frame transformations needed to achieve maximum signal convergence, it inherently targeted the local frame orientations used by the OMC system, which we knew to also be error-prone (due to inaccurate palpation, marker placement, etc.). For example, in some cases, the OMC-based abduction/adduction signals were clearly dependent on flexion angle, a strong indicator of cross-talk (c.f. knees 10, 29, and 32 in Appendix A). Moreover, this REFRAME configuration was not independent. In order to optimize one signal set, it relied on information contained within the other set. Consequently, we then implemented REFRAME_*RMS*_, seeking to achieve a similar level of signal convergence using a strictly self-contained method. Mean RMSEs between OMC on harness and the IMUs on harness decreased after REFRAME_*RMS*_, as did RMSEs between the OMC on harness and the IMUs on skin, although the latter did so to a lesser extent. Moreover, frame transformations implemented as part of REFRAME_*RMS*_ ranged from 0.0° to 32.7° (Appendix A). The results of this second REFRAME implementation therefore corroborated our previous findings; inconsistencies in reference frame orientations could account for most, but not all, differences between systems, especially between IMUs on skin and the optical reference on harness.

Even after REFRAME implementation, certain differences between signals remained, most obviously between external/internal rotation signals (as reflected by RMSE values) of IMUs on skin and OMC on harness (and IMUs on harness as well) (c.f. knee 11 in Appendix A). A tendency to estimate larger ranges of axial rotation with the IMUs on skin was observed, although it remained unclear whether this reflected an overestimation of external/internal rotation by the IMUs on skin vs. an underestimation of this rotation by the harness-mounted sensors. Moreover, signals from harness-mounted systems often displayed characteristic fluctuations, sometimes throughout the entire cycle (c.f. knee 22 in Appendix A), or specifically upon heel strike for the OMC system (c.f. knee 17 in Figure 3). Association with heel strike led us to hypothesize that the observed fluctuations could be due to the added mass of the harness, which interestingly was originally intended to reduce such artefacts [14]. From a simplified spring–mass system perspective, heel strike can be thought of as representing a transformation of kinetic energy into spring potential, resulting in vibration of the harness, much like a mass on a spring. A larger mass would logically lead to a larger relative displacement, plausibly explaining the differences in relative motion between sensors on skin and sensors on harness. Notably, although the general movement patterns after REFRAME were essentially the same for both harness-mounted systems, heel-strike-related fluctuations were visible mostly on OMC on harness signals only (and not on IMUs on harness signals). We hypothesized that this effect was likely associated with the use of Rauch–Tung–Striebel smoothing to estimate the knee joint angles from inertial data. As this method considered the levels of uncertainty associated with state variables at different timepoints, it was likely able to “reject” sensor measurements pointing to non-physiological movement patterns. Nevertheless, it remained unclear whether this smoothing effect could inadvertently discard real, relevant information (e.g., the presence of tremors in Parkinson’s patients).

The added mass of the harness was not the only concern associated with the OMC system. The brace was designed to act as a femoral “clamp”, where the lateral femoral attachment was meant to fit between the iliotibial band and the biceps femoralis tendon [14]. This not only led to difficulties related to harness placement in some of the more athletic participants (for example, due to muscle size obstructing this “groove”), but also to reported participant discomfort during dynamic activities and therefore possible gait alterations of natural movement patterns [40]. Although the system was only tested on a healthy population within this study, these effects would likely be exacerbated in a patient population displaying pathological patterns of gait. Notably, the technician supervising data collection mentioned noticing a visible reduction in flexion/extension range of motion between treadmill familiarization trials without the brace and data collection trials with the brace for several participants. We hypothesized that this could be associated with the lateral femoral attachment of the brace obstructing the motion of the iliotibial band, which would otherwise naturally move from being anterior to the lateral femoral epicondyle in full knee extension, to being posterior of the epicondyle with flexion [41]. Moreover, the optical marker-based system used in this study required the use of a treadmill for level walking to ensure that all markers stayed within the cameras’ field of view during the entire activity. Although this did allow for controlling of gait speed, it was nevertheless yet another limitation of the optical system that could be tackled using inertial sensors. The optical system also specifically instructed users to walk on the treadmill with socks for “better visualization”, which could be considered by some to be suboptimal (vs. walking in athletic shoes or barefoot).

In addition to participant discomfort arising from the use of the rigid harness and other usability-related issues that could clearly be improved, further limitations were identified. In a recent study, we established post-processing methods to ensure that consistent reference frame poses would ideally achieve three objectives: (1) clinical interpretability, (2) consistent reference frame orientation and position regardless of initial frame choice, and (3) method independence [11]. We concluded, however, that achieving all three was extremely challenging. REFRAME_*RMS*_ prioritized method independence to fulfill objective #3. In order to achieve a similar level of convergence as with REFRAME_*IMU*→*OMC*_ (and therefore attempt to fulfill objective #2), it was necessary to additionally target a common starting flexion angle of 0°. The comparably higher RMSE achieved in flexion/extension after REFRAME_*RMS*_ vs. REFRAME_*IMU*→*OMC*_, however, suggested that this objective was only partially fulfilled. Moreover, the target of 0° flexion at heel strike was chosen arbitrarily and was kept consistent across trials and participants for convenience. Fulfilling the remaining goal of clinical interpretability (objective #1) would naturally require targeting a clinically accurate value instead, determined based on, e.g., a sagittal view standing X-ray (assuming that knee flexion angle at heel strike was consistent with knee flexion angle during neutral standing). Likewise, the minimization of abduction/adduction RMS as part of REFRAME_*RMS*_ favored a zero mean for that signal, possibly leading to the loss of information regarding, e.g., static varus/valgus alignment. Importantly, while our study systematically analyzed agreement between the optical and inertial systems, establishing which of the systems most accurately captured the true motion of the underlying knee joints was considered a philosophical question beyond the scope of this investigation. The present comparison against the harness-based optical marker system was meant strictly to put the IMU-based estimates into context, by comparing against an established system that is currently used by experts for gait analysis [21,22,23]. It was certainly not meant to be an assessment of objective accuracy of the IMU-based system, as that would have required the use of, e.g., fluoroscopy to obtain soft-tissue-artefact-free kinematic measurements.

In conclusion, the negligible magnitude of the differences between the harness-mounted IMUs and the optical system after REFRAME conclusively demonstrated that the inertial-based system was capable of capturing the same underlying motion pattern as the optical-based system. On the other hand, the visible differences that remained between the skin-mounted IMUs and the optical reference indicated that the movement profile captured by the sensors on the harness was fundamentally different to that captured by IMUs on the skin, even after accounting for differences due to reference frame misalignment. This difference, however, did not conclusively indicate that the IMUs attached to the skin were subject to greater soft tissue artefact. In fact, the small fluctuations observed in the kinematic signals obtained from both the optical and inertial sensors that were fixed to the harness were suggestive of vibrations undergone by the rigid device upon heel strike, likely due to inertial effects resulting from its additional mass. Our study results propose that (1) the use of optical markers and camera systems can be successfully replaced by more cost-effective IMUs with similar accuracy (although further testing should more thoroughly assess performance in characterizing more complex activities and, e.g., pathological gait patterns), while (2) further investigation (especially in vivo and upon heel strike) against moving videofluoroscopy is recommended. Further testing should enable us to not only conclusively validate IMU-based knee kinematics, but also establish exactly how the kinematics captured using a rigid brace compare to the actual relative movement of the underlying bone segments.

## Figures and Tables

**Figure 1 bioengineering-11-00976-f001:**
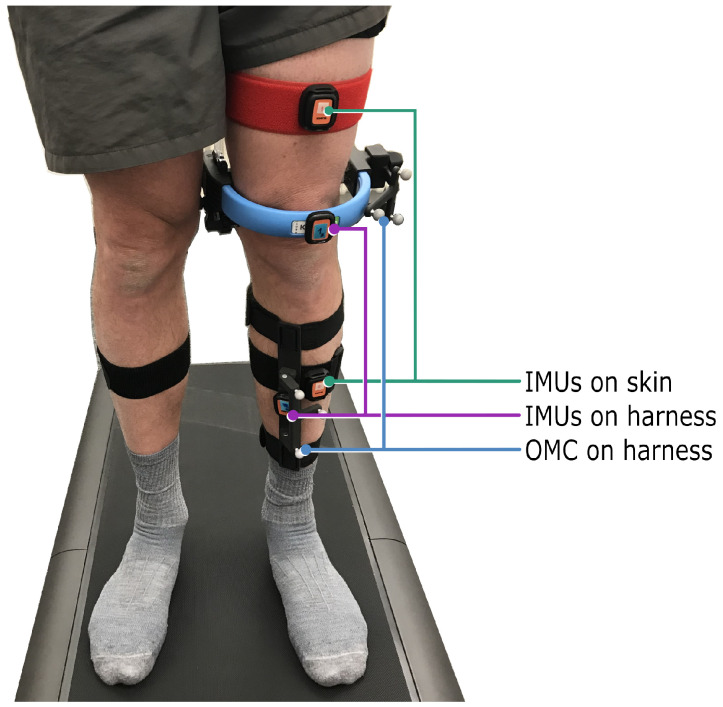
The optical harness-based reference system, as well as two pairs of IMU sensors, were carefully positioned on each participant by a certified technician. One IMU pair was attached to the rigid harness of the reference system (“IMUs on harness”), and a second IMU pair was mounted on elastic hook-and-loop bands (“IMUs on skin”). As per the optical system manufacturer’s instructions, participants walked in socks on the treadmill.

**Figure 2 bioengineering-11-00976-f002:**
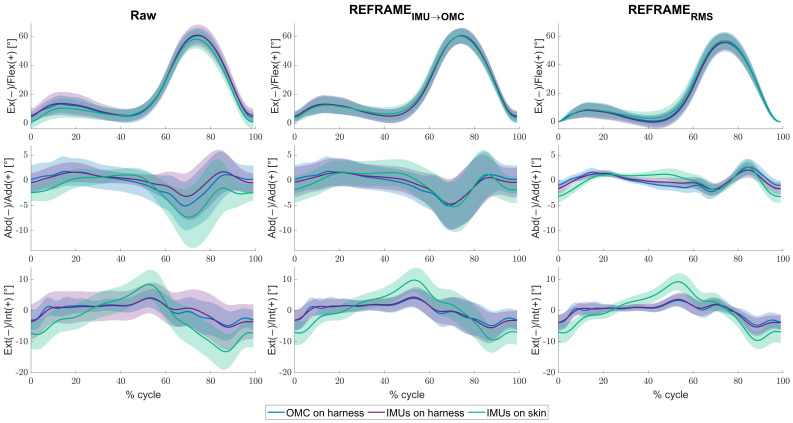
Mean tibiofemoral joint angles (solid lines) ± standard deviation (shaded areas), in degrees, as estimated by inertial measurement units (IMUs) on harness (purple), IMUs on skin (green), and optical motion capture (OMC) on harness (blue), averaged over all knees and cycles. Note that flexion angles have been illustrated as positive (despite representing a negative rotation around the laterally directed X-axis) for easier comparisons against other studies. Angles are shown as a percentage of the gait cycle under three conditions: (1) raw, i.e., in the absence of post-processing methods to correct reference frame orientation differences (**left**), (2) after implementation of REFRAME_*IMU*→*OMC*_ (**middle**), and (3) after implementation of REFRAME_*RMS*_ (**right**).

**Figure 3 bioengineering-11-00976-f003:**
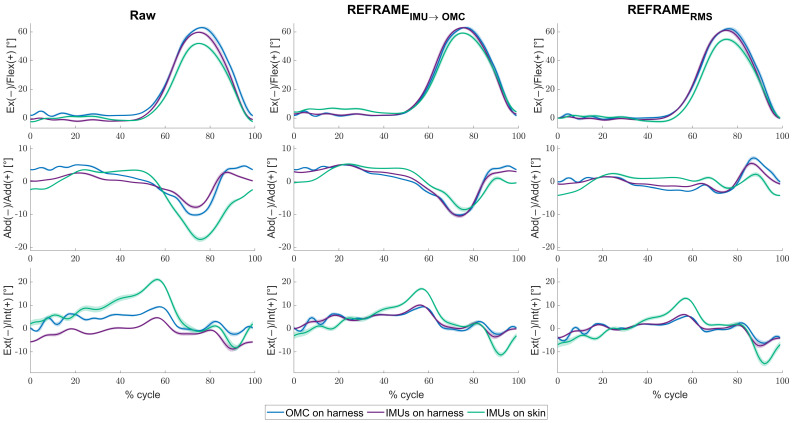
Mean tibiofemoral joint angles (solid lines) ± standard deviation (shaded areas), in degrees, as estimated by inertial measurement units (IMUs) on harness (purple), IMUs on skin (green), and optical motion capture (OMC) on harness (blue), averaged over all cycles for knee 17. Note that flexion angles have been illustrated as positive (despite representing a negative rotation around the laterally directed X-axis) for easier comparisons against other studies. Angles are shown as a percentage of the gait cycle under three conditions: (1) raw, i.e., in the absence of post-processing methods to correct reference frame orientation differences (**left**), (2) after implementation of REFRAME_*IMU*→*OMC*_ (**middle**), and (3) after implementation of REFRAME_*RMS*_ (**right**).

**Figure 4 bioengineering-11-00976-f004:**
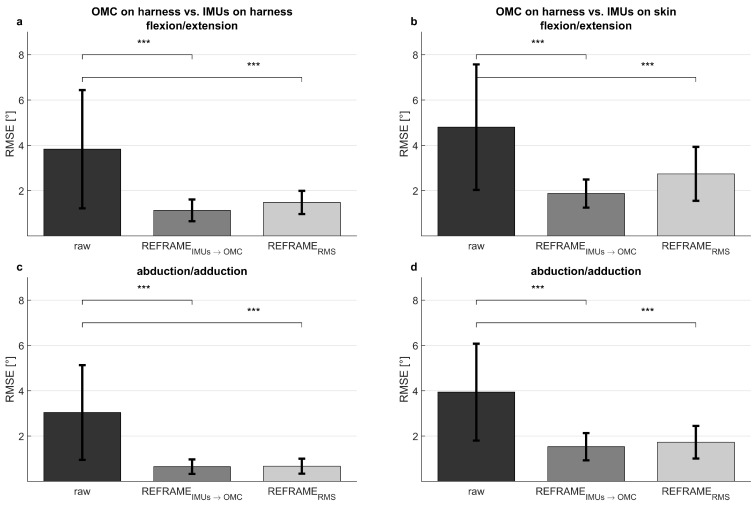
Mean ± standard deviation of root-mean-square errors (RMSEs, in degrees) between the optical reference system on a harness and the inertial measurement units on the harness (**left**), as well as between the optical reference system on a harness and the inertial measurement units on the skin (**right**). Shown for flexion/extension (**a**,**b**), abduction/adduction (**c**,**d**), and external/internal rotation (**e**,**f**). Significant changes in RMSEs after implementation of REFRAME_*IMU*→*OMC*_ and of REFRAME_*RMS*_, as determined by paired *t*-tests, are shown (*p* < 0.004 indicated by ***; full *p*-values are available in Appendix A).

**Table 1 bioengineering-11-00976-t001:** Inclusion and exclusion criteria for participation in the gait analysis study.

Inclusion Criteria	Exclusion Criteria
Employment at Aesculap AG in Tuttlingen	Impairment due to lower extremity, spine, or pelvic injury, or neurological disease
Ability to walk independently on a treadmill for five minutes at a time	Last surgical procedure less than six months ago
BMI ≤ 40 $\frac{\mathrm{kg}}{m^{2}}$	Pregnancy

**Table 2 bioengineering-11-00976-t002:** Mean ± standard deviation (in degrees) root-mean-square errors between tibiofemoral joint rotations estimated by the IMU-based systems and the optical reference on harness, calculated across all knees and cycles.

	Raw		REFRAME_*IMU*→*OMC*_		REFRAME_*RMS*_	
	IMUs on Harness	IMUs on Skin	IMUs on Harness	IMUs on Skin	IMUs on Harness	IMUs on Skin
flexion/extension	3.8 ± 2.6	4.8 ± 2.8	1.1 ± 0.5	1.9 ± 0.6	1.5 ± 0.5	2.7 ± 1.2
abduction/adduction	3.0 ± 2.1	3.9 ± 2.1	0.6 ± 0.3	1.5 ± 0.6	0.7 ± 0.3	1.7 ± 0.7
external/internal rotation	5.1 ± 2.7	6.5 ± 2.5	0.9 ± 0.3	4.1 ± 1.2	0.9 ± 0.3	4.0 ± 1.1

**Table 3 bioengineering-11-00976-t003:** Mean ± standard deviation (in degrees) root-mean-square errors between tibiofemoral joint rotations estimated by the IMU-based systems and the optical reference on harness, calculated across all cycles of an exemplary knee (knee 17).

	Raw		REFRAME_*IMU*→*OMC*_		REFRAME_*RMS*_	
	IMUs on Harness	IMUs on Skin	IMUs on Harness	IMUs on Skin	IMUs on Harness	IMUs on Skin
flexion/extension	4.1 ± 0.1	6.9 ± 0.1	1.7 ± 0.2	3.2 ± 0.1	2.0 ± 0.2	4.6 ± 0.2
abduction/adduction	2.4 ± 0.1	5.7 ± 0.1	0.8 ± 0.0	2.8 ± 0.1	1.0 ± 0.0	3.2 ± 0.1
external/internal rotation	5.3 ± 0.2	5.6 ± 0.3	1.0 ± 0.1	4.3 ± 0.2	1.0 ± 0.1	4.2 ± 0.2

**Table 4 bioengineering-11-00976-t004:** Mean ± standard deviation (in degrees) of the rotational transformations (Rx: rotation around X, Ry: rotation around Y, Rz: rotation around Z) applied to the femoral and tibial segment frames as part of REFRAME_*IMU*→*OMC*_, calculated across all knees and cycles.

	Femur		Tibia	
	IMUs on Harness	IMUs on Skin	IMUs on Harness	IMUs on Skin
Rx	0.0 ± 0.0	0.0 ± 0.0	0.5 ± 4.3	−1.9 ± 4.7
Ry	0.1 ± 2.8	−5.8 ± 4.9	0.7 ± 3.0	−5.5 ± 4.9
Rz	−2.4 ± 6.4	5.0 ± 7.9	−2.0 ± 8.4	5.0 ± 7.9

**Table 5 bioengineering-11-00976-t005:** Mean ± standard deviation (in degrees) of the rotational transformations (Rx: rotation around X, Ry: rotation around Y, Rz: rotation around Z) applied to the femoral and tibial segment frames as part of REFRAME_*IMU*→*OMC*_, calculated across all cycles of an exemplary knee (knee 17).

	Femur		Tibia	
	IMUs on Harness	IMUs on Skin	IMUs on Harness	IMUs on Skin
Rx	0.0 ± 0.0	0.0 ± 0.0	−3.7 ± 0.1	−6.7 ± 0.2
Ry	−0.5 ± 0.2	−4.7 ± 1.0	2.2 ± 0.2	−3.2 ± 1.0
Rz	−6.0 ± 0.2	11.3 ± 0.4	−0.4 ± 0.2	6.1 ± 0.3

**Table 6 bioengineering-11-00976-t006:** Mean ± standard deviation (in degrees) of the rotational transformations (Rx: rotation around X, Ry: rotation around Y, Rz: rotation around Z) applied to the femoral and tibial segment frames as part of REFRAME_*RMS*_, calculated across all knees and cycles.

	Femur			Tibia		
	OMC on Harness	IMUs on Harness	IMUs on Skin	OMC on Harness	IMUs on Harness	IMUs on Skin
Rx	0.0 ± 0.0	0.0 ± 0.0	0.0 ± 0.0	4.3 ± 3.9	5.5 ± 5.2	2.1 ± 5.0
Ry	−0.4 ± 5.3	−0.3 ± 5.6	−4.2 ± 7.0	−1.4 ± 4.3	−1.1 ± 5.4	−4.6 ± 6.8
Rz	5.1 ± 6.4	2.8 ± 4.7	9.3 ± 7.1	4.5 ± 6.7	2.5 ± 7.1	9.7 ± 6.5

**Table 7 bioengineering-11-00976-t007:** Mean ± standard deviation (in degrees) of the rotational transformations (Rx: rotation around X, Ry: rotation around Y, Rz: rotation around Z) applied to the femoral and tibial segment frames as part of REFRAME_*RMS*_, calculated across all cycles of an exemplary knee (knee 17).

	Femur			Tibia		
	OMC on Harness	IMUs on Harness	IMUs on Skin	OMC on Harness	IMUs on Harness	IMUs on Skin
Rx	0.0 ± 0.0	0.0 ± 0.0	0.0 ± 0.0	1.8 ± 0.6	−0.9 ± 0.5	−1.0 ± 0.6
Ry	1.7 ± 0.5	2.0 ± 0.5	−3.2 ± 1.5	−2.0 ± 0.5	1.0 ± 0.4	−4.6 ± 1.5
Rz	11.1 ± 0.3	5.3 ± 0.3	22.6 ± 0.5	7.0 ± 0.3	7.0 ± 0.3	13.9 ± 0.3

## Data Availability

The raw data supporting the conclusions of this article will be made available by the authors upon reasonable request.

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
