# Peer review of "Comparison of IMU-Based Knee Kinematics with and without Harness Fixation against an Optical Marker-Based System"

_bioengineering, 2024, doi:10.3390/bioengineering11100976_

Round 1

Reviewer 1 Report

Comments and Suggestions for Authors

This paper proposed a comparison study between optical makers and IMU sensors to demonstrate their effectiveness in motion encoding. The authors claim that IMU performs best as compared to optical markers to encode motion information.

The proposed study is explained in a good manner, however, I have the following major concerns:

1) IMU sensors have been used in several applications since the last decade to encode motion information as in [1,2,3]. The author must state their scientific contributions. In my opinion, the proposed study lacks novelty.

[1] Ashfaq, Nazish et al. "Identification of Optimal Data Augmentation Techniques for Multimodal Time-Series Sensory Data: A Framework." Information 15.6 (2024): 343.

[2] Batool, Sheeza et al. "An ensemble deep learning model for human activity analysis using wearable sensory data." Applied Soft Computing (2024): 111599.

[3] Fatima, Rimsha, et al. "A Systematic Evaluation of Feature Encoding Techniques for Gait Analysis Using Multimodal Sensory Data." Sensors 24.1 (2023): 75.

[4] Amjad, Fatima, et al. "A comparative study of feature selection approaches for human activity recognition using multimodal sensory data." Sensors 21.7 (2021): 2368.

2) The authors did not state how the ground truth is generated to measure the loss/error in the computation of motion information.

3) The authors must use similar criteria as in [5] to assess the loss/error.

[5] Khan, Muhammad Hassan, et al. "Marker-based movement analysis of human body parts in therapeutic procedure." Sensors 20.11 (2020): 3312.

4) The experiments are too limited and much needs to be enhanced.

5) Rather than attaching these sensors on these sensors (which may cause discomfort to humans), the authors need to use daily wearable devices to encode such information.

Comments on the Quality of English Language

Minor editing of English language required.

Author Response

For research article: “Comparison of IMU-Based Knee Kinematics with and without Harness Fixation against an Optical Marker-Based System

Response to Reviewer 1 Comments

1. Summary

Thank you very much for taking the time to review this manuscript and providing valuable feedback. Please find the detailed responses below and the corresponding revisions highlighted in yellow in the pdf version of the re-submitted files, as well as replicated below for the reviewer’s convenience.

2. Point-by-point response to Comments and Suggestions for Authors

Note to the reviewer:

Reference numbers within direct quotes correspond to the numbers used within the manuscript. Reference numbers used within our direct answers to the reviewer (i.e. outside of quotation marks) are specific to the reference list given at the end of this document.

Comment 1: IMU sensors have been used in several applications since the last decade to encode motion information as in [1,2,3]. The author must state their scientific contributions. In my opinion, the proposed study lacks novelty.

[1] Ashfaq, Nazish et al. "Identification of Optimal Data Augmentation Techniques for Multimodal Time-Series Sensory Data: A Framework." Information 15.6 (2024): 343.

[2] Batool, Sheeza et al. "An ensemble deep learning model for human activity analysis using wearable sensory data." Applied Soft Computing (2024): 111599.

[3] Fatima, Rimsha, et al. "A Systematic Evaluation of Feature Encoding Techniques for Gait Analysis Using Multimodal Sensory Data." Sensors 24.1 (2023): 75.

[4] Amjad, Fatima, et al. "A comparative study of feature selection approaches for human activity recognition using multimodal sensory data." Sensors 21.7 (2021): 2368.

Response 1: Thank you for pointing out that our scientific contribution could be stated more obviously and clearly. To address this issue, we have now added the following statement in lines 91-95, clearly delineating our scientific contribution and the novel aspects of our study:

“Previous studies implementing the REFRAME approach on IMU data tested either on a robotic joint simulator [9] or on cadaveric specimens [11]. This is thus the first study to account for potential differences in local segment reference frame orientations using an optimisation-based approach such as REFRAME within an in vivo evaluation of IMU-based tibiofemoral kinematics against optical motion capture.

Comment 2: The authors did not state how the ground truth is generated to measure the loss/error in the computation of motion information.

Response 2: Thank you very much for this comment. It is not fully clear to us exactly what the reviewer is referring to with “ground truth” or “loss/error”.

We purposefully avoid the use of the term “ground truth” throughout our manuscript as neither of the technologies assessed can really be considered “ground truth”. It is our understanding that the term “ground truth” refers to a “fundamental truth” or the “real” or “true” value of whatever signal is being approximated. In the context of tibiofemoral kinematics, a real ground truth could therefore only be obtained using e.g. videofluoroscopy or bone pins. Given the invasiveness of both methods, marker-based optical motion capture is instead often used as a reference for the assessment of new gait analysis technologies. Any such optical motion capture system, however, will inherently be affected by soft-tissue artefact, as it will not be able to directly track the movements of the underlying bones. Instead, tibiofemoral movements can only be approximated based on the movement of the visible outer limb segments (covered by muscle, ligaments, tendons, fat, skin, etc.). We therefore purposefully refer to the harness-based optical motion capture system as a “reference”. The harness-based system is considered an acceptable reference given its use in multiple previous studies [1, 2], in clinical settings [3,4], and its availability as a “portable, validated, FDA (510k) Cleared, Health Canada Licensed and CE marked commercial product” [5]. This idea has been highlighted in lines 409-414:

The present comparison against the harness-based optical marker system is meant strictly to put the IMU-based estimates into context, by comparing against an established system that is currently used by experts for gait analysis [21–23]. It is certainly not meant to be an assessment of objective accuracy of the IMU-based system, as that would require the use of e.g. fluoroscopy to obtain soft-tissue-artefact-free kinematic measurements.

If by “ground truth” the reviewer was referring to the KneeKG reference system, we can certainly more clearly direct the reader to the relevant literature where they can find the exact computational models used to estimate joint axes and angles with this device [6]. We have therefore made the following additions in lines 111-113:

“The computational method employed to define bone-embedded anatomical frames from marker positions and thus estimate knee joint angles has been previously described in detail by Hagemeister et al. [15].”

Likewise, we can also more clearly direct the reader to the literature describing the specific computational models used here to determine joint angles from inertial data using sensor fusion. These additions have been made to complement the already provided details in lines 165-172 (additions in yellow):

The underlying sensor fusion algorithm used for analysis of the IMU data was based on Ortigas-Vásquez et al.’s adaptation [9] of the approaches previously developed by Seel et al. [3] and Versteyhe et al. [17], which leverage Rauch-Tung-Striebel smoothing [26]. For further details the reader is referred to [3,9,17]. This allowed calculation of knee joint angles from the raw data sampled by each pair of IMUs, using custom Matlab scripts (vR2021b; The Mathworks Inc., Natick, Massachusetts, USA). In contrast, the OMC system used built-in software to automatically output joint angle estimates following an approach developed by the manufacturer [15].”

Moreover, we would like to highlight the information that had already been provided in lines 287-293, which already (at least partially) describes the methods used by each system to determine joint angles, while also citing the relevant literature where readers could obtain the full details if desired:

“As described by Hagemeister et al., the optical reference system leveraged joint axes that involved some level of functional calibration, but were still largely dependent on the manual identification of the 3D coordinates of key anatomical landmarks in a laboratory reference frame [15]. On the other hand, the joint axes implemented by the IMU-based system were based only on the available gyroscope and accelerometer data (as the position/orientation of bony points relative to the sensor positions/orientations were not directly measured) and were thus entirely functional [3, 9, 17].”

Regarding “loss/error”, we assume the reviewer is referring to the quantification of disagreement between the tibiofemoral kinematics estimated using the IMU-based system vs.  the harness-based optical system. This “error” has been quantified by calculating the root-mean-square errors (RMSEs). RMSE has been used in countless previous studies comparing measurement systems (e.g. [7-9]) and has long been an accepted measure of agreement between signals; More recently, the International Society of Biomechanics (ISB) has recommended the use of root-mean square error to indicate the magnitude of error when comparing an IMU-based system to a reference system [10].

Comment 3: The authors must use similar criteria as in [5] to assess the loss/error.

[5] Khan, Muhammad Hassan, et al. "Marker-based movement analysis of human body parts in therapeutic procedure." Sensors 20.11 (2020): 3312.

Response 3: Thank you for the input. Upon review of the cited literature, we established two key metrics were used in the mentioned article: 1) Average Joint Position Error (AJPE) and 2) Average Error in Angle Estimation. Rather than an analytical evaluation of the underlying sensor fusion algorithm implemented for joint angle estimation (which was performed previously within a separate study [11]), the aim of the present study is to compare the agreement between the IMU-based system in two configurations and a harness-based optical motion capture system acting as reference. AJPE is therefore not applicable in our study as the IMU-based system is not a visual tracking system; it does not directly measure the position of markers or joint centres. Due to the lack of positional information measured by the system, a criterion such as AJPE expressed in mm is thus not a suitable option here.

Average Error in Angle Estimation is indeed an appropriate criterion, but one that could be considered analogous/comparable to RMSE. There are several possible metrics to quantify agreement between signals, such as mean squared error, mean absolute error and root-mean-squared error. There is no objectively “best” choice, but when choosing one of them users usually consider factors such as robustness to outliers and interpretability. Some of the advantages that led us to choose RMSE for this study included interpretability (same units as input) and sensitivity to large errors (due to the square term) and thus its ability to emphasise notable discrepancies. We have previously used RMSE in multiple previous studies [11-14], as have several other research groups in related studies (e.g. [7-9]), allowing us to make easy comparisons. Moreover, as mentioned above, the ISB has indicated that root-mean square error is an appropriate metric to assess the concurrent validity of IMU-based systems against a reference system [10].

Comment 4: The experiments are too limited and much needs to be enhanced.

Response 4: Thank you for the feedback. We would be eager to enhance our experiments to continue further improving the quality of our research. It would therefore be really helpful if the reviewer could explicitly state what aspects of the study they consider to be limited and how it could be enhanced. For context, these preliminary in vivo tests will be followed by concurrent validation of the IMU-based system in vivo in 10 healthy subjects using optical skin-markers and moving dual-plane fluoroscopy.

Comment 5: Rather than attaching these sensors on these sensors (which may cause discomfort to humans), the authors need to use daily wearable devices to encode such information.

Response 5: Thank you for this comment. We appreciate the reviewer’s opinion on this point. We agree that in terms of usability, user-friendliness, comfort, etc., the harness-based system especially is not optimal. However, systematically assessing user comfort, etc., of different gait analysis systems was not an objective of this study. Our study aimed to assess the agreement between the tibiofemoral rotation signals output by two motion capture modalities, before and after REFRAME implementation.

Nevertheless, we fully appreciate that convenience and user comfort are certainly relevant aspects when evaluating gait analysis technologies. Despite not being the focus of this study, we therefore do still try to bring them to the reader’s attention multiple times in the discussion section, by repeatedly highlight the limitations of using a harness-based system, e.g. in lines 366- 387 (additions in yellow):

“The added mass of the harness was not the only concern associated with the OMC system. The brace was designed to act as a femoral “clamp”, where the lateral femoral attachment was meant to fit between the iliotibial band and the biceps femoralis tendon [14]. This not only led to difficulties related to harness placement in some of the more athletic subjects (for example due to muscle size obstructing this “groove”), but also to reported subject discomfort during dynamic activities and therefore possible gait alterations of natural movement patterns [37]. Although the system was only tested on a healthy population within this study, these effects would likely be exacerbated in a patient population displaying pathological patterns of gait. Notably, the technician supervising data collection mentioned noticing a visible reduction in flexion/extension range of motion between treadmill familiarisation trials without the brace and data collection trials with the brace for several subjects. We hypothesise this could be associated with the lateral femoral attachment of the brace obstructing the motion of the iliotibial band, which would otherwise naturally move from being anterior to the lateral femoral epicondyle in full knee extension, to be posterior of the epicondyle with flexion [38]. Moreover, the optical marker-based system used in this study requires the use of a treadmill for level walking to ensure all markers stay within the cameras’ field of view during the entire activity. Although this does allow for controlling of gait speed, it is nevertheless yet another limitation of the optical system that could be tackled using inertial sensors. The optical system also specifically instructed users to walk on the treadmill with socks for “better visualisation”, which could be considered by some to be suboptimal (vs. walking in athletic shoes or barefoot).”

To stress this point even further, we have now also added the following comment in lines 388-389 (additions highlighted in yellow):

“In addition to subject discomfort arising from the use of the rigid harness and other usability related issues that could clearly be improved, further limitations were identified.”

Reference List

[1] Clément, J.; de Guise, J.A.; Fuentes, A.; Hagemeister, N. Comparison of soft tissue artifact and its effects on knee kinematics between non-obese and obese subjects performing a squatting activity recorded using an exoskeleton. Gait & posture 2018, 61, 197–203. https://doi.org/10.1016/j.gaitpost.2018.01.009

[2] Gasparutto, X.; Bonnefoy-Mazure, A.; Attias, M.; Dumas, R.; Armand, S.; Miozzari, H. Comparison between passive knee kinematics during surgery and active knee kinematics during walking: A preliminary study. PloS one 2023, 18, e0282517. https://doi.org/10.1371/journal.pone.0282517

[3] Bytyqi, D.; Shabani, B.; Lustig, S.; Cheze, L.; Karahoda Gjurgjeala, N.; Neyret, P. Gait knee kinematic alterations in medial osteoarthritis: three dimensional assessment. International orthopaedics 2014, 38, 1191–1198. https://doi.org/10.1007/s00264-014-2312-3

[4] Cagnin, A.; Choinière, M.; Bureau, N.J.; Durand, M.; Mezghani, N.; Gaudreault, N.; Hagemeister, N. A multi-arm cluster randomized clinical trial of the use of knee kinesiography in the management of osteoarthritis patients in a primary care setting. Postgraduate medicine 2020, 132, 91–101. https://doi.org/10.1080/00325481.2019.1665457

[5] Emovi. (2023, March 25) Knee Kinesiography, a Dynamic Knee Exam | KNEEKG®. https://emovi.ca/

[6] Hagemeister, N.; Parent, G.; van de Putte, M.; St-Onge, N.; Duval, N.; de Guise, J. A reproducible method for studying three-dimensional knee kinematics. Journal of biomechanics 2005, 38, 1926–1931. https://doi.org/10.1016/j.jbiomech.2005.05.013

[7] Seel, T.; Raisch, J.; Schauer, T. IMU-based joint angle measurement for gait analysis. Sensors (Basel, Switzerland) 2014, 14, 6891–6909. https://doi.org/10.3390/s140406891 

[8] Zhou, L.; Tunca, C.; Fischer, E.; Brahms, C.M.; Ersoy, C.; Granacher, U.; Arnrich, B., Eds. Validation of an IMU Gait Analysis Algorithm for Gait Monitoring in Daily Life Situations: July 20-24, 2020 via the EMBS Virtual Academy; Vol. 2020, IEEE: Piscataway, NJ, 2020. https://doi.org/10.1109/EMBC44109.2020.9176827

[9] McGrath, T.; Stirling, L. Body-Worn IMU-Based Human Hip and Knee Kinematics Estimation during Treadmill Walking. Sensors (Basel, Switzerland) 2022, 22. https://doi.org/10.3390/s22072544

[10] Cereatti, A.; Gurchiek, R.; Mündermann, A.; Fantozzi, S.; Horak, F.; Delp, S.; Aminian, K. ISB recommendations on the definition, estimation, and reporting of joint kinematics in human motion analysis applications using wearable inertial measurement technology. Journal of biomechanics 2024, 173, 112225. https://doi.org/10.1016/j.jbiomech.2024.112225

[11] Ortigas-Vásquez, A.; Maas, A.; List, R.; Schütz, P.; Taylor, W.R.; Grupp, T.M. A Framework for Analytical Validation of Inertial-Sensor-Based Knee Kinematics Using a Six-Degrees-of-Freedom Joint Simulator. Sensors (Basel, Switzerland) 2022, 23. https://doi.org/10.3390/s23010348

[12] Ortigas-Vásquez, A.; Taylor, W.R.; Maas, A.; Woiczinski, M.; Grupp, T.M.; Sauer, A. A frame orientation optimisation method for consistent interpretation of kinematic signals. Scientific reports 2023, 13, 9632. https://doi.org/10.1038/s41598-023-36625-z

[13] Ortigas-Vásquez, A.; Taylor, W.R.; Postolka, B.; Schütz, P.; Maas, A.; Woiczinski, M.; Sauer, A. A Reproducible and Robust Representation of Tibiofemoral Kinematics of the Healthy Knee Joint during Stair Descent using REFRAME – Part I: REFRAME Foundations and Validation. Preprint on Research Square 2024. https://doi.org/10.21203/rs.3.rs-4207485/v1

[14] Sagasser, S.; Sauer, A.; Thorwächter, C.; Weber, J.G.; Maas, A.; Woiczinski, M.; Grupp, T.M.; Ortigas-Vásquez, A. Validation of Inertial-Measurement-Unit-Based Ex Vivo Knee Kinematics during a Loaded Squat before and after Reference-Frame-Orientation Optimisation. Sensors 2024, 24, 3324. https://doi.org/10.3390/s24113324

Reviewer 2 Report

Comments and Suggestions for Authors

This study investigated using IMUs as a practical alternative to optical marker systems for gait analysis in clinical setting by comparing knee joint angles from IMUs (mounted on either a harness or the skin) with those from optical markers. The authors applied the REFerence FRame Alignment MEthod (REFRAME), and obtained improved results for harness mounted IMUs, closely aligning with optical measurements, while skin-mounted IMUs showed less accuracy. Thus harness-mounted IMUs could potentially replace optical systems in clinical settings.

Among its strengths, the paper is clearly written and includes careful data analysis.

Among its weakness, the study hypothesis assumes validity of OMC for treadmill walking, which may be lacking.

1. The IMUs mounted on skin are more correctly described as mounted on elastic bands (line 8, Fig. 1).

2. The study aim is decribed as 'analytical validation of the described IMU-based knee kinematics analysis system under the possible influence of STA ' (line 73-74). However, the study methods, particularly OMC, do not analyze the effect of STA on OMC data.

3. The IMU data is compared against OMC data using KneeKG (line 80), however, by authors' own admission, KneeKG has not been validated on treadmill walking (line110).

4.  The reference cited for REFRAME method [18] on (line 85) appears incorrect. The REFRAME method was instead decribed in [11].

5. The range of walking speed on (line 98) appears erronous.

6. Does the second implementation of REFRAME on (line 170) involve applying REFRAME to the OMC signal as well? Wouldn't that jeopardize the study hypothesis?

7. Please report the p-values obtained in the t-tests mentions on (line 189).

8. How were OMC fluctuations on heel strike compentated (line 209)?

9. Please explain why REFRAME seems to be more effective for IMU on harness as compared to IMU on skin (Fig. 4).

10. In order to get a validation for the clinical standard, different walking pattern may be incorporated in the study as reported by [5]. 

11. Beyond STA, further investigation is needed to determine the remaining error variations between signals from harness-mounted configurations and the skin-mounted IMU.

Author Response

For research article: “Comparison of IMU-Based Knee Kinematics with and without Harness Fixation against an Optical Marker-Based System”

Response to Reviewer 2 Comments

  1. Summary

Thank you very much for taking the time to review this manuscript and providing valuable feedback. Please find the detailed responses below and the corresponding revisions highlighted in yellow in the pdf version of the re-submitted files, as well as replicated below for the reviewer’s convenience.

  1. Point-by-point response to Comments and Suggestions for Authors

Note to the reviewer:

Reference numbers within direct quotes correspond to the numbers used within the manuscript. Reference numbers used within our direct answers to the reviewer (i.e. outside of quotation marks) are specific to the reference list given at the end of this document.

Comment 1: The IMUs mounted on skin are more correctly described as mounted on elastic bands (line 8, Fig. 1).

Response 1: Thank you for pointing this out! We fully agree with the reviewer that making reference to the use of elastic bands here is a much better description of this IMU configuration. Our decision to refer to this configuration as “IMUs on skin” was to emphasise the lack of a rigid harness. To clarify this point, we have made updates to the manuscript accordingly. In lines 8-9, “IMUs mounted on the skin” are now:

“IMUs mounted on the skin using elastic hook-and-loop bands (from here on referred to as “skin-mounted IMUs”).”

Similar changes have been incorporated e.g. in lines 76-81:

“In the following study, we present an in vivo examination of IMU-based tibiofemoral kinematic estimates, considering two distinct configurations of IMU placement: 1) IMUs attached to a rigid harness (referred to as "IMUs on harness", i.e. with harness fixation), and 2) IMUs mounted "on the skin" using elastic hook-and-loop bands (referred to as "IMUs on skin", i.e. without harness fixation). (Note that "IMUs on skin" are technically not directly adhered onto the skin).”

Comment 2: The study aim is decribed as 'analytical validation of the described IMU-based knee kinematics analysis system under the possible influence of STA' (line 73-74). However, the study methods, particularly OMC, do not analyze the effect of STA on OMC data.

Response 2: Thank you very much for bringing this to our attention. It seems that our choice of words was not communicating the intended meaning. The reviewer is right to point out that our study does not systematically analyse the effect of STA on OMC (or IMU) data. Our intention was to point out that in both previous testing scenarios of our IMU-based system (on a robotic joint simulator [1], and in cadaveric specimen tests [2]) kinematic errors associated with soft-tissue artefact were not a possibility. Assessing the IMU-based system in a setup where soft-tissue artefact can be present is therefore an important next step to evaluate performance of the system in a more realistic context. Rather than quantify the exact effects of STA on IMU (or OMC data), which would require access to e.g. bone pin or fluoroscopy data, our goal was to first test the same IMU-based system in a scenario where the introduction of STA was possible to begin with. The question raised by the reviewer is nevertheless a valuable one, and one which we are in fact aiming to address in one of our next studies involving the simultaneous capture of in vivo tibiofemoral kinematics using OMC, IMU and dual-plane fluoroscopy data. We have now reworded the original statement to hopefully make our intended meaning more clear (see lines 74-76):

“Consequently, a next step in the analytical validation of the described IMU-based knee kinematics analysis system, is therefore to utilise the system in vivo, where results may be affected by errors due to STA.”

Comment 3: The IMU data is compared against OMC data using KneeKG (line 80), however, by authors' own admission, KneeKG has not been validated on treadmill walking (line 110).

Response 3: Definitely! This is an important limitation that we also identified and aimed to convey to the reader. For context, the KneeKG device was the OMC system that was made available to us for this series of tests. The system has previously been used in several prior studies [3,4] and even clinical settings (for treadmill walking) [5,6], under claims that it provides “repeatable and reliable knee kinematics” [7] and that it is a “valid […] and reliable tool…” [3]. We were only able to find evidence supporting these statements to a certain extent, which (like the reviewer pointed out) we aimed to highlight in lines 115-128:

“Flexion/extension, abduction/adduction and external/internal rotation values measured by the system have been previously validated at discrete intervals of knee flexion during a quasi-static weight-bearing squatting activity against radiographic images [16]. To the authors’ knowledge, a comprehensive validation of the system during treadmill walking (the system’s intended use) against fluoroscopic imaging is not yet available in the literature. Although the extent to which the device minimises STA during level walking has therefore yet to be directly assessed, multiple studies that utilise the optical reference system as a validated clinical gait analysis system or “silver standard” have been previously published [21–23]. Further studies by e.g. Lustig et al. [20], Clement et al. [16], and Northon et al. [24] commenting on the validity of the KneeKG system are also available for review. Mean repeatability values have similarly been reported by Hagemeister et al. [15] as ranging between 0.4° and 0.8° for joint rotations, although these values are expected to be highly optimistic and representative of a best-case scenario.”

Accordingly, we purposefully avoid labelling the OMC kinematics as “ground truth”, instead using the term “reference.” Furthermore, we now made the following additions in lines 406-414 to emphasise this point (additions in yellow):

Importantly, while our study systematically analyses agreement between the optical and inertial systems, establishing which of the systems most accurately captures the true motion of the underlying knee joints is considered a philosophical question that lies beyond the scope of this investigation. The present comparison against the harness-based optical marker system is meant strictly to put the IMU-based estimates into context, by comparing against an established system that is currently used by experts for gait analysis [21–23]. It is certainly not meant to be an assessment of objective accuracy of the IMU-based system, as that would require the use of e.g. fluoroscopy to obtain soft-tissue-artefact-free kinematic measurements.

Comment 4: The reference cited for REFRAME method [18] on (line 85) appears incorrect. The REFRAME method was instead decribed in [11].

Response 4: Thank you for bringing this to our attention. The preliminary method to REFRAME (FOOM), which addressed only frame orientations during flexion-dominant gait activities and only considered a limited set of fixed parameters to define the optimisation criteria was first described in [8 in this document; cited in the manuscript and by the reviewer as 18]. The full REFRAME method was then detailed in [9], currently available as a preprint. We have now added a citation to [9 in this document; cited in the manuscript as 19] in line 88.  The reference mentioned by the reviewer ([2 in this document: cited in the manuscript and by the reviewer as 11]) is actually a more recent investigation that applies REFRAME in the context of simulated squatting on cadaveric specimen tests.

Comment 5: The range of walking speed on (line 98) appears erronous

Response 5: Thank you for pointing this out! The previous values were a copy/paste error. We have now corrected this mistake in line 105, as well as thoroughly confirmed all other averages and standard deviations.

Comment 6: Does the second implementation of REFRAME on (line 170) involve applying REFRAME to the OMC signal as well? Wouldn't that jeopardize the study hypothesis?

Response 6: Yes and no. The reviewer is correct that the second implementation of REFRAME involves optimising the OMC signal as well, but this actually does not jeopardise the study hypothesis! The second REFRAME implementation seeks to achieve the same objective of reaching consistent reference frame orientations as the first REFRAME implementation but independently (without leveraging information contained within the OMC version of the dataset). In order to achieve this, rather than optimise one dataset towards the other (i.e. IMU towards OMC, which would be a dependent version of REFRAME), each version of the dataset has to be optimised entirely on its own. It is true that this means the orientations of the OMC reference frames within the femur and tibia segments will also be modified by REFRAME, but because these transformations are constant across the entire activity cycle, this has no effect on the underlying 3D physical motion between the segments. All that changes is the illustration of the OMC kinematics, while still representing the same underlying movement. Any independent REFRAME analysis will require every dataset being compared to be optimised on its own. Failing to do so would mean the frame orientations of whichever dataset were not optimised cannot be assumed to match that of those which have been optimised. To make this point clear we have added the following clarification in lines 188-190:

“Note that since frame transformations applied by REFRAME are constant across the entire activity cycle, the relative motion between limb segments actually remains the same (it is just illustrated differently).”

Comment 7: Please report the p-values obtained in the t-tests mentions on (line 189).

Response 7: Thank you for the feedback. The full p-values can now be found in section 3 of the Supplementary Material, in Tables S121 and S122, replicated below for the reviewer’s convenience:

Table S121: p-values associated with a two-tailed paired t-test performed on the mean RMSEs between tibiofemoral joint rotations estimated by the IMUs on harness and the OMC on harness.

Raw vs. REFRAMEIMUàOMC

Raw vs. REFRAMERMS

flexion/extension

1.3*10-7

2.9*10-6

abduction/adduction

9.0*10-9

1.0*10-9

external/internal rotation

1.5*10-12

1.4*10-12

Table S122: p-values associated with a two-tailed paired t-test performed on the mean RMSEs between tibiofemoral joint rotations estimated by the IMUs on skin and the OMC on harness.

Raw vs. REFRAMEIMUàOMC

Raw vs. REFRAMERMS

flexion/extension

1.2*10-7

4.4*10-5

abduction/adduction

1.7*10-9

7.6*10-9

external/internal rotation

6.7*10-8

2.0*10-8

Moreover, readers are now directed to refer to these p-values in the caption of Figure 4:

“Mean ± standard deviation of root-mean-square errors (RMSEs, in degrees) between the optical reference system on a harness and the inertial measurement units on the harness (left), as well as between the optical reference system on a harness and the inertial measurement units on the skin (right). Shown for flexion/extension (a, b), abduction/adduction (c, d), and external/internal rotation (e, f). Significant changes in RMSEs after implementation of REFRAMEIMUàOMC and of REFRAMERMS are as determined by paired t-tests are shown (p < 0.004 indicated by ***; full p-values are available in Supplementary Material Tables S121 and S122).”

As well as in line 252:

“Notably, all changes in mean RMSEs were found to be statistically significant even after Bonferroni correction (Figure 4; Supplementary Material Tables S121 and Table S122).”

Comment 8: How were OMC fluctuations on heel strike compentated (line 209)?

Response 8: Upon review, the text the reviewer seems to be referring to is the following in lines 225-229 (in the latest version of the document):

“Furthermore, OMC on harness kinematic signals showed clear fluctuations upon heel strike for several knees (e.g. Figure 3; for more examples see Supplementary Material). These fluctuations appeared highly repeatable, as demonstrated by the relatively small standard deviation across the trials (e.g. Table 3, for more examples see Supplementary Material).”

We are unfortunately not clear on what the reviewer is asking about here. Fluctuations on heel strike were not compensated by the OMC system; they were clearly visible in the OMC signals for several of the subjects, which is exactly what we were trying to point out in these lines.

Comment 9: Please explain why REFRAME seems to be more effective for IMU on harness as compared to IMU on skin (Fig. 4).

Response 9: Thank you for raising this question. REFRAME can only “improve” signal convergence (if that is possible between two specific sets of kinematic signals at all) by varying reference frame orientation and/or position (in this study, only the fixed orientations of the femoral and tibial frames). Signal convergence is optimised in such a way that any differences remaining after REFRAME could not possibly stem from differences in frame orientation and are therefore likely actual differences between the joint motion patterns quantified by the two sets of signals being compared. REFRAME does not change the underlying joint motion. REFRAME “seems to be more effective for IMU on harness as compared to IMU on skin” because both OMC on harness and IMU on harness are effectively quantifying the motion of the same rigid harness. Both the optical markers and the IMU sensors on the harness are effectively rigid relative to each other, so after frame alignment the signals are highly similar.  In contrast, the IMUs on skin are rather quantifying the motion of the elastic hook-and-loop bands wrapped tightly around the limb segments. Since the harness likely moves relative to the elastic hook-and-loop bands, even after reference frame alignment, the IMUs on skin and IMUs/OMC on harness do not quantify the same underlying movement pattern and so the signals remain visibly different.

To better communicate these ideas to the reader, we have now made additions in lines 299-323 accordingly:

“In order to address differences in joint axis orientations, REFRAMEIMUàOMC was applied to the set of kinematic signals stemming from each of the two IMU configurations (on harness and on skin) in turn. The underlying goal of this first REFRAME implementation was to re-align the IMU-based local segment reference frames to minimise the RMSE between IMU-based joint angles against the optical reference. The required transformations ranged from as little as 0.0° to as much as 31.0° (Supplementary Table S1). In addition to a visible improvement in signal convergence, average RMSEs decreased to well below 2° for the IMUs on harness (Table 2). The level of agreement observed after optimisation suggested the underlying motion captured by both systems was highly comparable.  Both the optical markers and the IMU sensors on the harness are effectively rigid relative to each other, so after frame alignment the resulting signals are highly similar. However, average RMSEs between OMC on harness and IMUs on skin after REFRAMEIMUàOMC were higher (up to 5°; Table 2), and clear disagreement remained between signals from the harness-mounted configurations and the skin-mounted IMU set up (Figure 2, middle column; Figure 3, middle column). The remaining differences could not plausibly stem from differences in frame alignment, so alternative sources of error, such as noise and/or measurement error, but especially differences in STA behaviour, are suspected. The differences in these kinematic signals therefore indicate that tibiofemoral motion as characterised by a harness-based system is inherently different to that captured by a system using skin-mounted sensors. In contrast to the IMUs on harness, the IMUs on skin quantify the motion of the elastic hook-and-loop bands wrapped tightly around the limb segments (rather than the motion of the harness). Since the harness likely moves relative to the elastic hook-and-loop bands, even after reference frame alignment, the IMUs on skin and IMUs on harness do not quantify the same underlying movement pattern and so those signals remain visibly different.”

Comment 10: In order to get a validation for the clinical standard, different walking pattern may be incorporated in the study as reported by [5].

Response 10: Thank you for this valuable input; we fully agree with the reviewer that further validation of the IMU-based system for clinical use should ideally involve variations of gait patterns (e.g. with different stride lengths as in Zhou et al. 2020), as well as activities other than level walking, such as stair descent/ascent, ramp ascent/descent, squat or sit-to-stand, etc. We will certainly incorporate these thoughts into the planning of our next set of experiments. We have now also commented on this idea in lines 426-431:

“Our study proposes that 1) the use of optical markers and camera systems can be successfully replaced by more cost-effective IMUs with similar accuracy (although further testing should more thoroughly assess performance in characterising more complex activities and e.g. pathological gait patterns), while 2) further investigation (especially in vivo and upon heel strike) against moving videofluoroscopy is recommended.”

Comment 11: Beyond STA, further investigation is needed to determine the remaining error variations between signals from harness-mounted configurations and the skin-mounted IMU.

Response 11: Certainly, we agree with the reviewer that differences between harness-mounted and skin-mounted kinematic signals are likely resulting from more than just STA. Signal differences demonstrate the harness and the elastic bands move relative to each other during the gait cycle. Nevertheless, the question remains which of the two is “more rigid” relative to the underlying femur and tibia bones we are aiming to measure, which should be further explored using e.g. videofluoroscopy, as suggested in lines 430-433:

“2) further investigation (especially in vivo and upon heel strike) against moving videofluoroscopy is recommended. Further testing should enable us to not only conclusively validate IMU-based knee kinematics, but also establish exactly how the kinematics captured using a rigid brace compare to the actual relative movement of the underlying bone segments.”

Reference List

[1] Ortigas-Vásquez, A.; Maas, A.; List, R.; Schütz, P.; Taylor, W.R.; Grupp, T.M. A Framework for Analytical Validation of Inertial-Sensor-Based Knee Kinematics Using a Six-Degrees-of-Freedom Joint Simulator. Sensors (Basel, Switzerland) 2022, 23. https://doi.org/10.3390/s23010348

[2] Sagasser, S.; Sauer, A.; Thorwächter, C.; Weber, J.G.; Maas, A.; Woiczinski, M.; Grupp, T.M.; Ortigas-Vásquez, A. Validation of Inertial-Measurement-Unit-Based Ex Vivo Knee Kinematics during a Loaded Squat before and after Reference-Frame-Orientation Optimisation. Sensors 2024, 24, 3324. https://doi.org/10.3390/s24113324

[3] Clément, J.; de Guise, J.A.; Fuentes, A.; Hagemeister, N. Comparison of soft tissue artifact and its effects on knee kinematics between non-obese and obese subjects performing a squatting activity recorded using an exoskeleton. Gait & posture 2018, 61, 197–203. https://doi.org/10.1016/j.gaitpost.2018.01.009

[4] Gasparutto, X.; Bonnefoy-Mazure, A.; Attias, M.; Dumas, R.; Armand, S.; Miozzari, H. Comparison between passive knee kinematics during surgery and active knee kinematics during walking: A preliminary study. PloS one 2023, 18, e0282517. https://doi.org/10.1371/journal.pone.0282517

[5] Bytyqi, D.; Shabani, B.; Lustig, S.; Cheze, L.; Karahoda Gjurgjeala, N.; Neyret, P. Gait knee kinematic alterations in medial osteoarthritis: three dimensional assessment. International orthopaedics 2014, 38, 1191–1198. https://doi.org/10.1007/s00264-014-2312-3

[6] Cagnin, A.; Choinière, M.; Bureau, N.J.; Durand, M.; Mezghani, N.; Gaudreault, N.; Hagemeister, N. A multi-arm cluster randomized clinical trial of the use of knee kinesiography in the management of osteoarthritis patients in a primary care setting. Postgraduate medicine 2020, 132, 91–101. https://doi.org/10.1080/00325481.2019.1665457

[7] Clément, J.; Dumas, R.; Hagemeister, N.; de Guise, J.A. Can generic knee joint models improve the measurement of osteoarthritic knee kinematics during squatting activity? Computer methods in biomechanics and biomedical engineering 2017, 20, 94–103. https://doi.org/10.1080/10255842.2016.1202935

[8] Ortigas-Vásquez, A.; Taylor, W.R.; Maas, A.; Woiczinski, M.; Grupp, T.M.; Sauer, A. A frame orientation optimisation method for consistent interpretation of kinematic signals. Preprint Scientific reports 2023, 13, 9632. https://doi.org/10.1038/s41598-023-36625-z

[9] Ortigas-Vásquez, A.; Taylor, W.R.; Postolka, B.; Schütz, P.; Maas, A.; Woiczinski, M.; Sauer, A. A Reproducible and Robust Representation of Tibiofemoral Kinematics of the Healthy Knee Joint during Stair Descent using REFRAME – Part I: REFRAME Foundations and Validation. Preprint on Research Square 2024. https://doi.org/10.21203/rs.3.rs-4207485/v1

Reviewer 3 Report

Comments and Suggestions for Authors

This paper compare the results of the knee motion during the treadmill walking obtained with 3 methods. This paper is an interesting one from methodological point of view, nevertheless there are some concernes connected with it, which should be addressed prior to its publication.

1. The IMU data was compared with the data obtained by optical system KneeKG. This is not a standard 3D optical system with well known human body model and validated marker set. Some information about this system should be given: its repeatability, validity, etc. This is especially important, as this system serves as "golden standard" for the IMU data.

2. All subjects were healthy adults. This is a limitation, as in patients the distorted movements of the body segments during gait could give bigger differences than in healthy subjects. It should be somehow discussed at the end of the paper.

3. There are two other limitations: treadmill walking, which is not the natural way the humans move, in case of clinical application in the future this also should be discussed. Maybe the use of the treadmill was due to easier way of comapring the results, but this should be clearly stated. Another one is shown on fig. 1 - socks. Were the subjects walking in socks? Or some in socks, some barefoot? In clinical gait analysis walking in socks is not recommended due to safety reasons.

4. The flexion/extension presentation on Fig. 2 is not the accepted standard of ISB: it should be mirrored upside down.

Author Response

For research article: “Comparison of IMU-Based Knee Kinematics with and without Harness Fixation against an Optical Marker-Based System

Response to Reviewer 3 Comments

1. Summary

Thank you very much for taking the time to review this manuscript and providing valuable feedback. Please find the detailed responses below and the corresponding revisions highlighted in yellow in the pdf version of the re-submitted files, as well as replicated below for the reviewer’s convenience.

2. Point-by-point response to Comments and Suggestions for Authors

Note to the reviewer:

Reference numbers within direct quotes correspond to the numbers used within the corresponding original manuscripts. Reference numbers used within our direct answers to the reviewer (i.e. outside of quotation marks) are specific to the reference list given at the end of this document.

Comment 1: The IMU data was compared with the data obtained by optical system KneeKG. This is not a standard 3D optical system with well known human body model and validated marker set. Some information about this system should be given: its repeatability, validity, etc. This is especially important, as this system serves as "golden standard" for the IMU data.

Response 1: Thank you for pointing this out. Previous studies have made the following statements regarding the KneeKG system’s accuracy and repeatability:

Source

Statement

Lustig et al. (2012) [1]

https://doi.org/10.1007/s00167-011-1867-4

“average accuracy of 0.4° of knee abduction and adduction, 2.3° for axial rotation, 2.4 mm for anteroposterior translation, and 1.1 mm for axial translation”

Lustig et al. (2012) [1]

https://doi.org/10.1007/s00167-011-1867-4

“The accuracy of this system was assessed by Hagemeister et al. [9].  On 16 healthy subjects, they found intra-patient reproducibility between 0.86 and 0.97 for abduction/adduction, internal/external rotation, and flexion/extension movements. In a different study, Hagemeister et al.  [10] determined the mean repeatability of measures to range between 0.4° and 0.8° for knee rotation angles and between 0.8 and 2.2 mm for translation.”

Clement et al. (2016) [2]

https://doi.org/10.1080/10255842.2016.1202935

“The exoskeleton provides a repeatable and reliable knee kinematics, but is still influenced by STA errors which can reach 7° and 11 mm for knee bone rotations and displacements (Südhoff et al. 2007).”

Clement et al. (2018) [3]

10.1016/j.gaitpost.2018.06.024

“The objective of the present study was to characterize 3D knee kinematics during gait in healthy women and men with a validated tool.”

Clement et al. (2018) [3]

10.1016/j.gaitpost.2018.06.024

“Our main objective was thus to describe natural 3D knee kinematics using a non-invasive, validated, and reliable tool, the KneeKG™ system, in a large cohort of healthy women and men.”

Clement et al. (2018) [3]

10.1016/j.gaitpost.2018.06.024

“The KneeKG is a valid [25] and reliable [[26], [27], [28]] tool that limits soft tissue artifact [29,30].”

Northon et al. (2018) [4]

10.1016/j.knee.2018.08.011

“has an accuracy of 0.8° and 2.4 mm for frontal plane and anteroposterior motions during functional tasks”

Nevertheless, we consider these reported values to be highly optimistic and not fully representative of what real errors from the KneeKG system are likely to be in practice. Instead of repeating these values, we therefore summarised the extent to which the KneeKG system has been previously validated in lines 115-123 as follows:

“Flexion/extension, abduction/adduction and external/internal rotation values measured by the system have been previously validated at discrete intervals of knee flexion during a quasi-static weight-bearing squatting activity against radiographic images [16]. To the authors’ knowledge, a comprehensive validation of the system during treadmill walking (the system’s intended use) against fluoroscopic imaging is not yet available in the literature. Although the extent to which the device minimises STA during level walking has therefore yet to be directly assessed, multiple studies that utilise the optical reference system as a validated clinical gait analysis system or “silver standard” have been previously published [21–23].”

To also point the reader towards more of these relevant studies that present the KneeKG as a validated system, we have now also added the following statement in lines 123-128, highlighted in yellow:

“Further studies by e.g. Lustig et al. [20], Clement et al. [16], and Northon et al. [24] commenting on the validity of the KneeKG system are also available for review. Mean repeatability values have similarly been reported by Hagemeister et al. [15] as ranging between 0.4° and 0.8° for joint rotations, although these values are expected to be highly optimistic and representative of a best-case scenario.”

Moreover, we highlight in lines 406-409 the fact that although the KneeKG system acts as a reference within the context of this study, it should not necessarily be considered more “accurate” than the IMU-based system:

“Importantly, while our study systematically analyses agreement between the optical and inertial systems, establishing which of the systems most accurately captures the true motion of the underlying knee joints is considered a philosophical question that lies beyond the scope of this investigation.”

Furthermore, to make this point clearer to the reader, we have now made the following additions, highlighted in yellow, in lines 409-414:

“The present comparison against the harness-based optical marker system is meant strictly to put the IMU-based estimates into context, by comparing against an established system that is currently used by experts for gait analysis [21–23]. It is certainly not meant to be an assessment of objective accuracy of the IMU-based system, as that would require the use of e.g. fluoroscopy to obtain soft-tissue-artefact-free kinematic measurements.”

Comment 2: All subjects were healthy adults. This is a limitation, as in patients the distorted movements of the body segments during gait could give bigger differences than in healthy subjects. It should be somehow discussed at the end of the paper.

Response 2: Thank you very much for bringing this to our attention. To highlight this issue as a limitation of the study we have made the following addition in lines 372-374:

“Although the system was only tested on a healthy population within this study, these effects would likely be exacerbated in a patient population displaying pathological patterns of gait.

Comment 3: There are two other limitations: treadmill walking, which is not the natural way the humans move, in case of clinical application in the future this also should be discussed. Maybe the use of the treadmill was due to easier way of comapring the results, but this should be clearly stated. Another one is shown on fig. 1 - socks. Were the subjects walking in socks? Or some in socks, some barefoot? In clinical gait analysis walking in socks is not recommended due to safety reasons.

Response 3: Thank you for the input. We will certainly take into account the safety issues associated with sock use into future studies. For now, we have added the following statements in lines 380-387 to better describe the limitations associated with treadmill use.

“Moreover, the optical marker-based system used in this study requires the use of a treadmill for level walking to ensure all markers stay within the cameras’ field of view during the entire activity. Although this does allow for controlling of gait speed, it is nevertheless yet another limitation of the optical system that could be tackled using inertial sensors. The optical system also specifically instructed users to walk on the treadmill with socks for “better visualisation”, which could be considered by some to be suboptimal (vs. walking in athletic shoes or barefoot).”

Comment 4: The flexion/extension presentation on Fig. 2 is not the accepted standard of ISB: it should be mirrored upside down.

Response 4: Thank you for the feedback regarding our choice to display extension as positive. We agree that from a clinical perspective, knee flexion is often illustrated as positive in kinematic plots. In our technical description of the directions of the coordinate system axes in lines 175-179 however, we clearly state that the X-axis points medially, the Y-axis points anteriorly and the Z-axis points proximally. From a mathematical perspective, as per the right-hand rule (we are using right-handed coordinate systems), a positive rotation of the tibial segment around the X-axis relative to the femur segment would be equivalent to an anticlockwise rotation, representing knee joint extension. Nevertheless, to align with ISB standards as specified by the reviewer, the figures have now been adapted to ensure flexion is shown as positive.

Reference List

[1] Lustig, S.; Magnussen, R.A.; Cheze, L.; Neyret, P. The KneeKG system: a review of the literature. Knee surgery, sports traumatology, arthroscopy: official journal of the ESSKA 2012, 20, 633–638. https://doi.org/10.1007/s00167-011-1867-4

[2] Clément, J.; Dumas, R.; Hagemeister, N.; de Guise, J.A. Can generic knee joint models improve the measurement of osteoarthritic knee kinematics during squatting activity? Computer methods in biomechanics and biomedical engineering 2017, 20, 94–103. https://doi.org/10.1080/10255842.2016.1202935

[3] Clément, J.; Toliopoulos, P.; Hagemeister, N.; Desmeules, F.; Fuentes, A.; Vendittoli, P.A. Healthy 3D knee kinematics during gait: differences between women and men, and correlation with x-ray alignment. Gait Posture 2018, 64, 198–204. https://doi.org/10.1016/j.gaitpost.2018.06.024

[4] Northon, S.; Boivin, K.; Laurencelle, L.; Hagemeister, N.; de Guise, J.A. Quantification of joint alignment and stability during a single leg stance task in a knee osteoarthritis cohort. The Knee 2018, 25, 1040–1050. https://doi.org/10.1016/j.knee.2018.08.011

Round 2

Reviewer 1 Report

Comments and Suggestions for Authors

I'm disappointed to note that the authors did not address the comments from the first revision. I still have the following major concerns:

  • The paper lacks novelty.
  • The experiments are too limited and require significant enhancement.

Additionally, please note that "ground truth" refers to the actual labeled data.

Comments on the Quality of English Language

Minor editing of English language required.

Author Response

We would like to thank the Reviewers for their time and effort in reviewing this manuscript and for their valuable feedback. As recommended by Assistant Editor Cary Liu, we have submitted a detailed rebuttal letter addressing Reviewer 1’s Round 2 comments on 12 September, in addition to the concerns we previously raised about Reviewer 1's reports in our email of 5 September. We would like to inform the Reviewer that the rebuttal letter has been accepted.

Reviewer 2 Report

Comments and Suggestions for Authors

The authors have satisfactorily answered my previous comments. I approve the current version for publication.

Author Response

We would like to thank the Reviewers for their time and effort in reviewing this manuscript and for their valuable feedback. We are pleased to hear that we have satisfactorily addressed the Reviewers' comments and that they approve the current version for publication.